# Age at Natural Menopause in Women Living with HIV: A Cross-Sectional Study Comparing Self-Reported and Biochemical Data

**DOI:** 10.3390/v15051058

**Published:** 2023-04-26

**Authors:** Shayda A. Swann, Elizabeth M. King, Shelly Tognazzini, Amber R. Campbell, Sofia L. A. Levy, Neora Pick, Jerilynn C. Prior, Chelsea Elwood, Mona Loutfy, Valerie Nicholson, Angela Kaida, Hélène C. F. Côté, Melanie C. M. Murray

**Affiliations:** 1Experimental Medicine, University of British Columbia, Vancouver, BC V5Z 1M9, Canada; sswann19@student.ubc.ca (S.A.S.);; 2Women’s Health Research Institute, Vancouver, BC V6H 3N1, Canada; 3Faculty of Health Sciences, Simon Fraser University, Burnaby, BC V5A 1S6, Canada; 4Oak Tree Clinic, BC Women’s Hospital and Health Centre, Vancouver, BC V6H 3N1, Canada; 5Department of Pathology and Laboratory Medicine, University of British Columbia, Vancouver, BC V5Z 1M9, Canada; 6Faculty of Science, University of British Columbia, Vancouver, BC V5Z 1M9, Canada; 7Centre for Menstrual Cycle and Ovulation Research, University of British Columbia, Vancouver, BC V5Z 1M9, Canada; 8School of Population and Public Health, University of British Columbia, Vancouver, BC V5Z 1M9, Canada; 9Department of Obstetrics and Gynecology, University of British Columbia, Vancouver, BC V5Z 1M9, Canada; 10Women’s College Research Institute, Women’s College Hospital, Toronto, ON M5G 1N8, Canada; 11Centre for Blood Research, University of British Columbia, Vancouver, BC V5Z 1M9, Canada; 12Edwin S.H. Leong Healthy Aging Program, University of University of British Columbia, Vancouver, BC V5Z 1M9, Canada; 13Department of Medicine, University of British Columbia, Vancouver, BC V5Z 1M9, Canada

**Keywords:** menopause age, HIV, women’s health, follicle-stimulating hormone, early menopause

## Abstract

Early menopause (<45 years) has significant impacts on bone, cardiovascular, and cognitive health. Several studies have suggested earlier menopause for women living with HIV; however, the current literature is limited by reliance on self-report data. We determined age at menopause in women living with HIV and socio-demographically similar HIV-negative women based on both self-report of menopause status (no menses for ≥12 months) and biochemical confirmation (defined as above plus follicle-stimulating hormone level ≥ 25 IU/mL). Multivariable median regression models assessed factors associated with menopause age, controlling for relevant confounders. Overall, 91 women living with HIV and 98 HIV-negative women were categorized as menopausal by self-report, compared to 83 and 92 by biochemical confirmation. Age at menopause did not differ significantly between groups, whether based on self-report (median [IQR]: 49.0 [45.3 to 53.0] vs. 50.0 [46.0 to 53.0] years; *p* = 0.28) or biochemical confirmation (50.0 [46.0 to 53.0] vs. 51.0 [46.0 to 53.0] years; *p* = 0.54). In the multivariable model, no HIV-related or psychosocial variables were associated with earlier age at menopause (all *p* > 0.05). Overall, HIV status per se was not statistically associated with an earlier age at menopause, emphasizing the importance of comparing socio-demographically similar women in reproductive health and HIV research.

## 1. Introduction

The prevalence of HIV among women and girls is increasing globally, comprising 54% of the 38.4 million people living with HIV in 2021 [1]. Concurrently, as people live longer with effective treatment, approximately 70% of people living with HIV are predicted to be over the age of 50 by 2030 in high-income countries [2]. Given this, it is increasingly salient to understand the menopause experience of women living with HIV. Menopause not only signals the end of reproductive years but may also bring bothersome vasomotor symptoms (i.e., hot flushes and night sweats) that could require lifestyle and/or pharmacological management [3]. Menopause also signals a need for increased bone and cardiovascular health monitoring [4,5].

The World Health Organization defines menopause as the lack of a menstrual period for at least 12 consecutive months [6]; thus, age at menopause is calculated as one year after the final menstruation. The current literature estimating age at menopause in women living with HIV is inconsistent, with studies reporting both earlier [7,8,9,10,11] and equivalent [12,13] ages in women living with HIV compared to their HIV-negative counterparts. Accurately determining whether women living with HIV experience menopause earlier in life than HIV-negative peers has important implications for fertility planning, as well as screening for and treating vasomotor symptoms, bone loss, cognitive dysfunction, and cardiovascular disease. Early menopause (age < 45 years) is associated with an increased risk of these adverse health effects [14,15,16], making this a priority area of consideration for women living with HIV.

The majority of studies of menopause age in women living with HIV determined menopause status by self-report [17]. However, relying solely on self-report to determine menopausal status may misclassify women who experience adaptive and reversible prolonged amenorrhea [18], particularly in groups with a high prevalence of menstrual irregularities, such as women with HIV [19]. These women have been reported to experience prolonged amenorrhea at a frequency of 20%, much higher than that of the general population [20], for reasons likely owing to their increased exposure to physiological and psychological stressors [21], opiate use [22,23], nutritional deficiency, or low body weight [18,24]. Furthermore, in one prospective US cohort study of 660 women living with HIV, more than one-third of women living with HIV experiencing amenorrhea had resumed menses within one year, demonstrating that many women with amenorrhea were not menopausal [25]. Hypothalamic amenorrhea can be readily distinguished from menopause using follicle-stimulating hormone (FSH) levels, which increase in menopause [26] but are low in hypothalamic amenorrhea [27]. Consequently, FSH may be a useful adjunct tool to differentiate amenorrhea from menopause and establish menopausal age.

Given this evidence, our aims were to (1) compare the reproductive phase (i.e., premenopausal or menopausal) by self-report and with additional biochemical confirmation with FSH levels and (2) determine and compare the average age at natural (non-surgical) menopause in women living with HIV compared with HIV-negative women sharing similar socio-demographic characteristics.

## 2. Materials and Methods

### 2.1. Study Design and Participants

This analysis included women enrolled from two prospective community cohorts studying aging in women living with HIV. The Children and Women: AntiRetroviral Therapy and Markers of Aging (CARMA) study enrolled > 1000 women living with and without HIV in British Columbia (BC) from 2008–2018, a subset of whom (*n* = 239) completed an endocrinology-focused questionnaire in the CARMA-ENDO sub-study from January 2013 to August 2017. The BC CARMA-CHIWOS (Canadian HIV Women’s Sexual and Reproductive Health Cohort Study) Collaboration (BCC3) is a continuation of the CARMA study that examines intersecting cellular, biomedical, and psychosocial-structural factors that impact healthy aging in women living with HIV and well-matched HIV-negative women [28]. Data collection for BCC3 began in December 2020 and is ongoing. Both studies intentionally enrolled women living with and without HIV in BC, Canada, who shared similar demographics and social identities. Data collection took place at the women- and family-centered Oak Tree Clinic at BC Women’s Hospital and other clinical sites. Both studies received approval from the University of British Columbia Children’s and Women’s Research Ethics Board (H09-02867 and H19-00896).

Participants were eligible for inclusion in this analysis if they were sex-assigned female at birth and ≥35 years old at the visit. Although we acknowledge that this definition does not include all people who identify as women nor all people born with ovaries, this analysis focuses on women who would have a biological experience of menopause. Women were excluded if they had experienced hysterectomy or bilateral oophorectomy, were currently taking hormonal contraceptives (including a levonorgestrel-releasing intrauterine system), or were currently using systemic menopause hormone therapy. If a participant was enrolled in both studies, only the CARMA data were considered to ensure distinct participants.

### 2.2. Procedures

Women living with HIV were recruited from the Oak Tree Clinic and local HIV/AIDS service organizations, through Craigslist ads, social media, and by word-of-mouth. HIV-negative women were recruited from the local community via social media and advertisements at clinics and community centers, with an attempt to enroll those of similar socio-economic status as those who were living with HIV. A detailed description of the BCC3 study protocol has been previously published [28]. Briefly, study visits included two detailed questionnaires (available at www.hivhearme.ca), the collection of biospecimens, and anthropometric measurements. Questionnaires and study procedures used in the CARMA study were very similar to those used in the BCC3 study. All participants provided written and informed consent at enrollment.

### 2.3. Variables

Demographic data included age, ethnicity, income, and education. Clinical variables consisted of body mass index (BMI, kg/m^2^), age at menarche, parity (number of live births), history of ever having hepatitis C virus infection, past or current tobacco smoking, and history of ever using illicit substances, including opioids, cocaine/crack, and methamphetamine. For women living with HIV, we also collected data on nadir and current CD4 count, highest ever and current HIV viral load, and years lived with HIV. FSH levels were measured in real-time from plasma at the BC Women’s Hospital Laboratory using the VITROS Immunodiagnostic Products FSH Reagent Pack, with an intra-assay coefficient of variation (CV) of 2.6% and inter-assay CV of 6.2% at 20.6 IU/mL (Ortho-Clinical Diagnostics, Rochester, NY, USA).

### 2.4. Defining Menopause Status and Age at Menopause

Menopause status was determined both by self-report of no flow for ≥12 months and with biochemical confirmation using FSH levels. For self-report, participants were asked how they would describe their current menstrual status as it relates to menopause, with options including premenopausal, perimenopausal, or menopausal. Women were provided with definitions for each of these terms with respect to time since their last menstrual period, cycle length/regularity, and menopause symptoms. Women reporting amenorrhea for at least 12 months, and either age ≥ 60 years or with FSH levels ≥ 25 IU/mL, were considered to have biochemically confirmed menopause [29]. In addition, women over the age of 55 with a BMI ≥ 35 kg/m^2^ and ≥12 months of amenorrhea were classified as menopausal regardless of FSH level, as women with a high BMI tend to have lower FSH (*n* = 4 participants) [30]. Finally, some women with past or current opioid use had low FSH levels despite clinical symptoms of menopause; these women were classified as menopausal and then removed from the sensitivity analysis, as described below. Age at menopause was calculated as one year following the last reported menstrual period.

### 2.5. Sample Size and Statistical Analysis

The sample size was calculated a priori to yield 90% power to detect a two-year difference in age at menopause between groups, as reported in other studies showing a difference in age at menopause [7,10]. Based on this, 84 women with biochemically confirmed menopause would be required in each group. Demographic and clinical variables were compared between women living with and without HIV using T, Mann–Whitney U, or chi-square tests, as appropriate for the data type and distribution. Concordance between self-reported and FSH-confirmation of menopause status was compared by a chi-squared test and assessed separately for women living with HIV and HIV-negative women. Age at menopause was compared between groups by the Mann–Whitney U test. Univariable and multivariable median regression models were constructed to determine factors associated with age at menopause, including HIV status, ethnicity, income, BMI, age at menarche, parity, substance use, tobacco smoking, and history of ever having hepatitis C virus infection. Variables were selected for inclusion in the final model based on Akaike Information Criteria values. For women living with HIV, we also investigated univariable associations between age at menopause and HIV-specific parameters. Univariable and multivariable logistic regression models were constructed to determine factors associated with the odds of early menopause, defined as <45 years. Similarly, we tested whether the proportion of women with primary ovarian insufficiency (POI, menopause age < 40 years) differed between groups. To reduce potential confounding by opioid use, we performed a post hoc sensitivity analysis whereby women with any history of opioid use were excluded. The significance level was α = 0.05 and all statistical analyses were completed using R version 4.2.2.

## 3. Results

### 3.1. Participant Demographics

Of the 622 women who completed CARMA or BCC3 study visits, 187 women living with HIV and 193 HIV-negative women met eligibility for inclusion in this analysis (Figure 1). The two groups differed with respect to ethnicity (*p* = 0.002), and women living with HIV were more likely to have a low income (*p* = 0.003), low education (*p* < 0.001), higher parity (*p* = 0.007), current or past hepatitis C virus infection (*p* < 0.001), current or past tobacco smoking (*p* = 0.004), and history of using substances (*p* = 0.003) (Table 1). Participants were similar in age, BMI, and age at menarche (all *p* > 0.05). Among women living with HIV, 83.8% had HIV viral loads < 40 copies/mL and 86.2% had a current CD4 count greater than 200 cells/mm^3^.

### 3.2. Classification of Reproductive Phase

Based on self-report of no menses for ≥12 months, 91 women living with HIV and 98 HIV-negative women reported being menopausal. After assessing FSH levels, 83 women living with HIV and 92 controls were classified as menopausal. Demographic and clinical characteristics of the menopausal participants are outlined in Table 2. Classification of menopause status did not differ between self-report and biochemical confirmation in either group (women living with HIV: *p* = 0.20; HIV-negative women: *p* = 0.35). Overall, there was concordance between self-report and biochemical confirmation of menopause in 92.5% of women living with HIV and 93.8% of HIV-negative women. The reproductive phase was misclassified by self-report in 14 women living with HIV and 12 HIV-negative women (Appendix A). Among those with discordance between self-report and biochemically confirmed reproductive phase, 19/26 (73.1%) had current or past substance use. Of the eleven women living with HIV who were misclassified as “menopausal” by self-report, six would have been considered to have POI (i.e., menopause before age 40) and three reported becoming menopausal at age 45.

### 3.3. Age at Menopause

Based on self-report data, the age at which women living with HIV reached menopause was not statistically different than that of HIV-negative women (median [IQR]: 49.0 [45.3 to 53.0] vs. 50.0 [46.0 to 53.0] years; *p* = 0.28, Figure 2). When we restricted this analysis to only include women with biochemically confirmed menopause, we continued to observe no significant difference between groups, and the median menopause age increased by one year for both women living with HIV and HIV-negative women (50.0 [46.0 to 53.0] vs. 51.0 [46.0 to 53.0] years; *p* = 0.54, Figure 2). One control participant who met our biochemical definition of menopause (46.9 years old and FSH level of 39.9 IU/mL) had a menopause age of 16 years. While it is unlikely that the participant truly became menopausal at this age, they were retained in the analysis as we are unable to estimate their age of final menstrual period otherwise. Excluding this participant had no bearing on the results.

In the univariate median regression analyses, current tobacco smoking was the only variable associated with lower age at menopause (β (95% CI): −3.00 (−4.95 to −1.05); *p* < 0.001) (Table 3). In the multivariable model, no variables were independently associated with menopause age (all *p* > 0.05; Table 3). Similarly, no HIV-related variables were associated with the odds of early menopause (all *p* > 0.05; Table 4).

We also tested the association between HIV status and the odds of early menopause. A similar proportion of women living with HIV and HIV-negative women experienced earlier menopause based on the biochemical definition (14/83, 16.9% vs. 15/92, 16.3%; *p* = 1.00). In the multivariable logistic regression model, we found no association between HIV status and the odds of early menopause (adjusted OR: 1.35 (0.47 to 3.88); *p* = 0.57) (Appendix A). Further, we found that 8/83 (9.64%) women living with HIV and 8/92 (8.70%) HIV-negative women had POI (*p* = 0.83).

### 3.4. Sensitivity Analyses

We performed a sensitivity analysis whereby women with past or current opioid use or who preferred not to answer substance use questions were excluded from the analysis. After applying this exclusion, 68 women living with HIV who were menopausal by self-report (66 by biochemical confirmation) and 86 HIV-negative women (85 by self-report) remained. Here, we found similar results, with no significant difference in age at menopause between women living with HIV and HIV-negative women based either on self-report (50.0 [46.0 to 53.0] vs. 51.0 [46.0 to 53.0] years; *p* = 0.58) or biochemical confirmation (50.0 [47.6 to 53.0] vs. 51.0 [46.0 to 53.0]; *p* = 0.92). When we repeated the multivariable regression, we continued to see no association between HIV status and age at menopause (Appendix A). In this model, current tobacco smoking was the only variable independently associated with earlier age at menopause (−2.77 (−5.14 to −0.41); *p* = 0.02).

## 4. Discussion

In this analysis, we observed no statistically significant difference in age at menopause between women living with HIV and HIV-negative women, based on both self-report and when FSH levels were considered. These results are reinforced by the finding that no HIV-specific variables were associated with age at menopause. Furthermore, we found that women living with HIV and HIV-negative women reported their current reproductive phases with high accuracy.

We presumed that the similar menopause ages observed in women living with HIV and controls in this study were driven by the health of participants, the majority of whom had healthy BMIs and CD4 counts; both factors that have previously been associated with age at menopause in HIV and general population studies [12,31]. Our results stand in contrast to other studies that have found a lower age at menopause among women living with HIV, which may be due to differences in study populations and analyses. For instance, four studies showing earlier menopause in women living with HIV included women in the general population as the control group [7,8,10,12], precluding the ability to compare to socio-demographically similar HIV-negative women. For instance, Boonyanurak et al. (2012) reported a mean menopause age of 47.3 among 55 women living with HIV in Thailand, compared to the general population average of 49.5 years [7]. In a Canadian study of 149 naturally menopausal women living with HIV, Andany et al. (2020) found a median menopause age of 49.0 years, compared to the population average of 51.0 years [10]. Importantly, Boonyanurak et al. (2012) highlighted the need for future studies to include an internal control group, as we have done [7]. To our knowledge, one study has previously enrolled an HIV-negative control group; however, our results may not be comparable, as only 65% of women in that study were using antiretrovirals [11]. In light of this, our results suggest that socio-demographic factors are the main drivers of age at menopause, as has been demonstrated previously in HIV-negative women [32]. That being said, we note that our results cannot exclude a potential one-year difference in age at menopause, which, although not statistically significant, may be clinically meaningful. Furthermore, age at menopause was lower than the general Canadian population average of 51.0 years in both groups [33], emphasizing the likely influence of shared socio-economic conditions.

To our knowledge, only one other study has classified menopause status using biochemical data [34]. In that cross-sectional analysis of 1449 women living with HIV (743 menopausal) in the United States, menopause status was defined as no menses for ≥12 months and anti-Mullerian hormone levels < 20 pg/mL. The authors reported the median age at menopause as 48.0 years among women who met this definition of menopause. In a modeling analysis including both premenopausal and menopausal women, the predicted median age at menopause was 50.0 years, very similar to what we have reported here (median [IQR] 50.0 [46.0 to 53.0] years). In a sensitivity analysis censoring women with detectable anti-Mullerian hormone levels, they reported a median menopause age of 49.0 years, which again falls within the range found in our results. Overall, our data are consistent with this existing study that utilizes biochemical data.

We observed that the self-reported age at menopause among women living with HIV was one year younger than that by FSH ≥ 25 IU/mL. In addition, we found that 11 women living with HIV reported being menopausal despite low FSH levels, suggesting hypothalamic amenorrhea. The majority of these women would have been classified as having early menopause or POI by self-report alone. Our results are comparable to those reported by Andany et al. (2020) [10], where 7.7% of women living with HIV in their Canadian cohort experienced spontaneous POI, compared to 9.6% in our cohort. One existing study reported a higher prevalence of POI in women living with HIV compared to an HIV-negative control group with menopause status defined by self-report [11]. In contrast, we found no difference in the prevalence of POI based on biochemical confirmation, suggesting that FSH levels may be particularly useful in these instances. These results emphasize the importance of including biochemical tests in women aged <40–45 years when clinically indicated (i.e., when menstrual periods have ceased for greater than one year), including FSH, luteinizing hormones, beta-hCG, prolactin, and thyroid stimulating hormones. In such cases, biochemical testing may identify reversible causes of amenorrhea. These findings also underscore the importance of discussing menstrual health and menopause with women living with HIV. Previous data indicated that only 44.8% of midlife women living with HIV in a Canadian cohort had discussed menopause with their healthcare providers [35]. Taken together, these results point to the importance of empowering women to understand their reproductive health as it changes throughout their life and highlight the need for advocacy from healthcare providers.

Overall, high proportions of both women living with HIV and HIV-negative women in our study reported experiencing early menopause or POI, with frequencies far exceeding those typically seen in the general population. Specifically, 16.5% of all women in our cohort met the criteria for early menopause and 9.1% for POI, much higher than the population averages of 7.6% and 2%, respectively [36,37]. We suspect this reflects socio-demographic risk factors for earlier menopause that are shared among women in our cohort, with a substantial proportion of participants experiencing low income and current/past tobacco smoking, both of which are associated with earlier menopause [33].

In our multivariable analysis, substance use was not independently associated with earlier age at menopause. In contrast to our findings, Schoenbaum et al. (2005) observed that heroin/cocaine use was independently associated with early menopause, with an OR of 2.63 (1.61 to 4.31) [11]. Our results may vary due to differences in the study population, as only 34% of menopausal women in our cohort had current or past substance use compared to 52% in the study by Schoenbaum et al. More data on midlife adults who use substances are required to adequately investigate this relationship.

This study is limited in that we did not investigate other causes of amenorrhea (e.g., thyroid abnormalities, hyperprolactinemia, structural causes, etc.). Furthermore, the cross-sectional nature of these data required that we determine age at the last menstrual period based on self-report, which may be impacted by recall bias. Similarly, we are unable to measure FSH levels at the onset of menopause and hence cannot confirm precisely when women became menopausal, which would require prospective data. This may have biased the estimate of concordance between self-report and biochemical data if women with previous hypothalamic amenorrhea had since reached natural menopause at the time of the study. We also note that FSH levels are an imperfect estimate of menopause status and can be impacted by numerous factors, including age, BMI, calorie intake, genetic variability, tobacco smoking, opioid use, and severe illness [22,38,39]. To mitigate the risk of misclassification, we incorporated several of these factors into our definition of biochemically confirmed menopause and in sensitivity analyses but acknowledge that unmeasured confounders may have influenced our categorization of women into reproductive phases. We also note that this analysis is not powered to detect less than a two-year difference in menopause age. Finally, the majority of women who were enrolled in our study came from the Oak Tree Clinic, which practices women-centered and specialized HIV care [40]. As such, most participants were medically well, and very few had detectable HIV viral loads or CD4 counts < 200 cells/mL; therefore, it is unknown whether our results would be translatable to women with limited access to high-quality HIV care.

This analysis also has several strengths. First, our robust dataset enables careful consideration of biomedical variables that may impact menopause age, such as BMI, age at menarche, and psychosocial variables that are often underreported in midlife women, such as substance use. Furthermore, the inclusion of a community-dwelling HIV-negative group facilitated direct comparisons to socio-demographically similar women. In addition, we enrolled sufficient numbers to have the power to show a two-year difference in menopause age. Lastly, utilizing a biochemical marker for menopause allowed us to rule out potentially confounding effects of hypothalamic amenorrhea.

## 5. Conclusions

Herein, we present unique, cross-sectional data to show no statistical difference in menopausal age in women living with and without HIV. These findings suggest that factors beyond HIV status likely contribute a greater role to age at menopause. In summary, this work indicates that women living with HIV who have access to high-quality healthcare may experience near-normal ages at menopause.

## Figures and Tables

**Figure 1 viruses-15-01058-f001:**
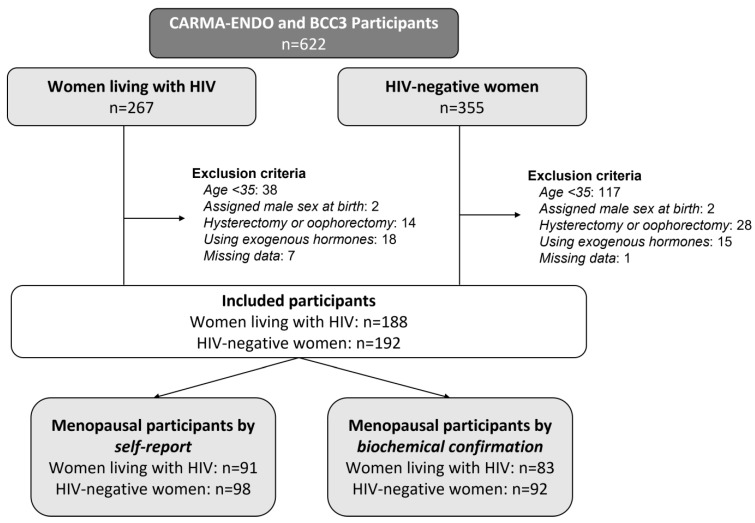
Participant selection by HIV status for comparison of age at menopause (≥12 months since final menstruation) in the CARMA-ENDO and BCC3 cohorts.

**Figure 2 viruses-15-01058-f002:**
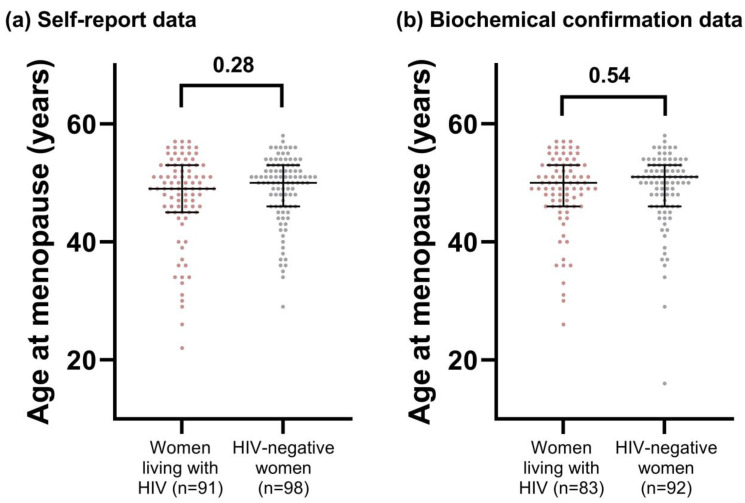
Comparing age at menopause in women living with and without HIV in the CARMA-EDNO and BCC3 cohorts by (**a**) self-report of no menses for ≥12 months and (**b**) no menses for ≥12 months with follicle-stimulating hormone levels ≥ 25 IU/mL (biochemical confirmation). Self-reported age at menopause is based on a clinical survey. Biochemical data is based on self-report with follicle-stimulating hormone levels. Bars represent the median and interquartile range. Median age at menopause was compared by the Mann–Whitney U test, with *p*-values indicated above the plots.

**Table 1 viruses-15-01058-t001:** Participant demographics, HIV status, and clinical characteristics in the investigation of menopause age in the CARMA-ENDO and BCC3 cohorts.

	Women Living with HIV (*n* = 188)	HIV-Negative Women (*n* = 192)	*p*-Value
**Socio-demographic data**			
Age, median [IQR] * years	49.2 [43.4 to 55.8]	51.0 [43.8 to 58.1]	0.16
Ethnicity, *n* (%)			0.002
White	75 (39.9)	79 (41.1)	
Indigenous	66 (35.1)	54 (28.1)	
African, Caribbean, or Black	26 (13.8)	13 (6.8)	
Other/mixed	21 (11.2)	46 (24.0)	
Income, *n* (%)			0.003
<CAD 20,000 †/year	92 (52.3)	66 (36.1)	
≥CAD 20,000/year	86 (46.7)	117 (63.9)	
Education, *n* (%)			<0.001
≤High school	98 (53.3)	51 (26.6)	
>High school	86 (46.7)	141 (73.4)	
**Clinical data**			
BMI ^‡^, median [IQR] kg/m^2^	26.7 [21.8 to 31.2]	26.5 [22.2 to 30.5]	0.83
Age at menarche, median [IQR] years	13 [12 to 14]	13 [12 to 14]	0.72
Parity, median [IQR]	2 [1 to 3]	1 [0 to 2]	0.007
Hepatitis C virus infection, *n* (%)			<0.001
Never	96 (51.3)	161 (84.3)	
Ever	91 (48.7)	30 (15.7)	
Tobacco smoking, *n* (%)			0.004
Never	67 (35.6)	89 (46.4)	
Past	33 (17.6)	45 (23.4)	
Current	88 (46.8)	58 (30.2)	
Substance use ^§^, *n* (%)			0.003
Never	101 (54.3)	122 (63.9)	
Past	41 (22.0)	49 (25.7)	
Current	44 (23.7)	20 (10.5)	
**HIV-related clinical data**			
Nadir CD4 count, *n* (%)		N/A ‖	N/A
<200 cells/mL	50 (53.2)		
200–500 cells/mL	40 (42.6)		
>500 cells/mL	4 (4.3)		
Current CD4 count, *n* (%)		N/A	N/A
<200 cells/mL	22 (13.8)		
200–500 cells/mL	42 (26.4)		
>500 cells/mL	95 (59.7)		
Current HIV viral load, *n* (%)		N/A	N/A
Undetectable (<40 copies/µL)	155 (83.8)		
Detectable (≥40 copies/µL)	30 (16.2)		
Highest ever HIV viral load, *n* (%)		N/A	N/A
<100,000 copies/mL	65 (46.4)		
≥100,000 copies/mL	75 (53.6)		
Years lived with HIV, median [IQR]	19 [12 to 23]	N/A	N/A

* IQR = interquartile range; † CAD = Canadian dollars; ‡ BMI = body mass index; § Substance use = past/present use of opioids, crack/cocaine, and/or methamphetamines. ‖ N/A = not applicable. Missing data: Income (*n* = 21), education (*n* = 4), BMI (*n* = 12), tobacco smoking (*n* = 1), hepatitis C virus infection (*n* = 2), substance use (*n* = 4), menarche (*n* = 6), nadir CD4 count (*n* = 92), current CD4 count (*n* = 28), HIV viral load (*n* = 2), years lived with HIV (*n* = 2).

**Table 2 viruses-15-01058-t002:** Demographics and clinical characteristics of menopausal participants by HIV status in the CARMA-ENDO and BCC3 cohorts.

	Menopausal by Self-Report (*n* = 189)			Menopausal by Self-Report with Biochemical Confirmation (*n* = 175)		
	Women Living with HIV (*n* = 91)	HIV-Negative Women (*n* = 98)	*p*-Value	Women Living with HIV (*n* = 83)	HIV-Negative Women (*n* = 92)	*p*-Value
**Socio-demographic characteristics**						
Age, median [IQR *] years	55.9 [52.2 to 59.1]	57.9 [54.0 to 62.9]	0.03	56.8 [53.7 to 59.5]	58.1 [53.9 to 62.9]	0.13
Ethnicity, *n* (%)			0.26			0.19
White	39 (42.9)	50 (51.0)		35 (42.2)	49 (53.3)	
Non-White	52 (57.1)	48 (49.0)		48 (57.8)	43 (46.7)	
Income, *n* (%)			0.001			<0.001
<CAD 20,000 †/year	51 (58.0)	30 (32.6)		46 (57.5)	26 (29.5)	
≥CAD 20,000/year	37 (42.0)	62 (67.4)		40 (49.4)	68 (73.9)	
Education, *n* (%)			0.002			0.003
≤High school	46 (52.3)	28 (28.6)		41 (50.6)	24 (26.1)	
>High school	42 (47.7)	70 (71.4)		40 (49.4)	68 (73.9)	
**Clinical characteristics**						
BMI ^‡^, median [IQR] kg/m^2^	25.7 [20.8 to 31.2]	26.0 [22.1 to 30.0]	0.33	26.6 [20.9 to 31.2]	26.4 [22.1 to 30.5]	0.34
Age at menarche, median [IQR] years	13 [12 to 14]	13 [12 to 14]	0.21	13 [12 to 14]	13 [12 to 14]	0.13
Age at menopause, median [IQR] years	49.0 [45.3 to 53.0]	50.0 [46.0 to 53.0]	0.28	50.0 [46.0 to 53.0]	51.0 [46.0 to 53.0]	0.54
Parity, median [IQR]	2 [1 to 3]	2 [0 to 3]	0.08	2 [1 to 3]	2 [0 to 2]	0.15
Hepatitis C virus infection, *n* (%)			<0.001			<0.001
Never	37 (40.7)	82 (83.7)		38 (45.8)	79 (85.9)	
Ever	54 (59.3)	16 (16.3)		45 (54.2)	13 (14.1)	
Tobacco smoking, *n* (%)			0.03			0.008
Never	33 (36.3)	46 (46.9)		32 (38.6)	46 (50.0)	
Past	16 (17.6)	25 (25.5)		15 (18.1)	26 (28.3)	
Current	42 (46.2)	27 (27.6)		36 (43.4)	20 (21.7)	
Substance use ^§^, *n* (%)			0.03			0.008
Never	53 (58.2)	63 (64.3)		51 (61.4)	62 (67.4)	
Past	17 (18.7)	26 (26.5)		15 (18.1)	25 (27.2)	
Current	21 (23.1)	9 (9.2)		17 (20.5)	5 (5.4)	
**HIV-related variables**						
Nadir CD4 count, *n* (%)		N/A ‖	N/A		N/A	N/A
<200 cells/mL	27 (56.2)			26 (60.5)		
200–500 cells/mL	20 (41.7)			17 (39.5)		
>500 cells/mL	1 (2.1)			0 (0)		
Current CD4 count, *n* (%)		N/A	N/A		N/A	N/A
<200 cells/mL	9 (11.2)			8 (10.8)		
200–500 cells/mL	23 (28.7)			21 (28.4)		
>500 cells/mL	48 (60.0)			45 (60.8)		
Current HIV viral load, *n* (%)		N/A	N/A		N/A	N/A
Undetectable (<40 copies/µL)	78 (85.7)			71 (85.5)		
Detectable (≥40 copies/µL)	13 (14.3)			12 (14.5)		
Highest HIV viral load, *n* (%)		N/A	N/A		N/A	N/A
<100,000 copies/µL	31 (45.6)			27 (45.0)		
≥100,000 copies/µL	37 (54.4)			33 (55.0)		
Years lived with HIV, median [IQR]	21 [15 to 25]	N/A	N/A	21 [15 to 25]	N/A	N/A

* IQR = interquartile range; † CAD = Canadian dollars; ‡ BMI = body mass index. § Substance use = past/current use of opioids, crack/cocaine, and/or methamphetamine. ‖ N/A = not applicable.

**Table 3 viruses-15-01058-t003:** Factors associated with self-reported and biochemically confirmed age at menopause in women living with HIV (*n* = 83) and HIV-negative women (*n* = 92) by multivariable median regression.

	Unadjusted β (95% CI)	*p*-Value	Adjusted β (95% CI)	*p*-Value
HIV status (ref: HIV-negative)	−1.00 (−3.04 to 1.04)	0.33	0.86 (−0.72 to 2.43)	0.28
Income (ref: ≥CAD 20,000/year)	−2.00 (−4.47 to 0.47)	0.11	0.05 (−1.66 to 1.77)	0.95
Parity (per increase in one birth)	0.00 (−0.71 to 0.71)	1.00	0.00 (−0.54 to 0.54)	0.99
Body mass index (per one kg/m^2^ increase)	−0.00 (−0.10 to 0.10)	1.00	−0.04 (−0.14 to 0.06)	0.41
Hepatitis C virus infection (ref: never)	−2.00 (−4.45 to 0.45)	0.11	−1.00 (−2.94 to 0.94)	0.31
Tobacco smoking (ref: never)				
Past	0.00 (−1.66 to 1.66)	1.00	−1.55 (−3.35 to 0.26)	0.09
Current	−3.00 (−4.95 to −1.05)	<0.001	−2.62 (−5.48 to 0.24)	0.07
Substance use * (ref: never)				
Past	−1.00 (−2.72 to 0.72)	0.25	−0.31 (−2.31 to 1.68)	0.76
Current	−4.00 (−8.96 to 0.96)	0.11	−3.14 (−8.11 to 1.82)	0.21
Ethnicity (ref: White)				
Non-White	−1.00 (−2.88 to 0.88)	0.29		
Age at menarche (per one-year increase)	−0.00 (−0.51 to 0.51)	1.00		

Biochemically confirmed age at menopause was determined based on the age at the last menstrual period and follicle-stimulating hormone levels. * Substance use = history of opioid, crack/cocaine, and/or methamphetamine use. Bolded values are statistically significant at *p* < 0.05.

**Table 4 viruses-15-01058-t004:** Association between HIV-related variables and self-reported and biochemically confirmed age at menopause among women living with HIV (*n* = 83).

	Unadjusted β (95% CI)	*p*-Value	Adjusted β (95% CI)	*p*-Value
Nadir CD4 count (ref: ≥200 cells/mL)				
<200 cells/mL	0.00 (−7.76 to 7.76)	1.00	0.00 (−5.33 to 5.33)	1.00
Current CD4 count (ref: ≥500 cells/mL)				
<500 cells/mL	−2.00 (−1.08 to 5.08)	0.20	3.00 (−1.85 to 7.85)	0.21
Highest HIV viral load (ref: <100,000 copies/µL)				
≥100,000 copies/mL	−1.00 (−4.39 to 2.39)	0.56	−2.00 (−9.33 to 5.33)	0.58
Current HIV viral load (ref: undetectable)				
Detectable	−2.50 (−5.99 to 0.99)	0.16		
Years lived with HIV (per one-year increase)	−0.08 (−0.27 to 0.11)	0.39		

Biochemically confirmed age at menopause was determined based on the age at the last menstrual period and follicle-stimulating hormone levels.

## Data Availability

Data are available upon request to the corresponding author.

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
