# Peer review of "Age at Natural Menopause in Women Living with HIV: A Cross-Sectional Study Comparing Self-Reported and Biochemical Data"

_viruses, 2023, doi:10.3390/v15051058_

Round 1

Reviewer 1 Report

The manuscript is focused on age at natural menopause in HIV-infected women. In this cross-sectional study, the authors performed two methods to define menopause which is reasonable. It is well written with sound English. Though the size was not large enough to find the difference between groups, the study group might be women living with HIV receiving suppressive ART, the authors carefully made the conclusion and gave some definitions to generalize the results, which is acceptable.

Author Response

We thank Reviewer 1 for the kind and thoughtful feedback on this manuscript.

Reviewer 2 Report

Dear Authors, 

This is an interesting and well written paper describing a study evaluating the age at menopause of women living with HIV by both self-report and biochemical measures with HIV uninfected women.  I believe this study which looked at FSH levels and had a well matched control group does add to the literature and will be of interest to those caring for women living with HIV.  

I found the article to be clear and easily understandable.  I am not a statistician but the statistical methods appear sound and easily understandable.  

I agree with the conclusions that in women well-matched that HIV alone does not appear to have a significant impact on the age of menopause.  

Author Response

We thank Reviewer 2 for an accurate overview and positive assessment of this manuscript.

Reviewer 3 Report

My doubt is on the layout of the manuscript, ie on the diagnosis of menopause: generally, when the FSH values are high, the 17 beta-estradiol values are low, we are in a situation of menopause.

If, on the other hand, there are higher FSH levels, higher than 30 mIU/ml, the menopause process has already begun, in particular, for example, if values higher than 40mIU/ml are found in two consecutive doses performed one month apart, fertility is now exhausted, and we find ourselves in the presence of an impending situation of menopause.

However, the FSH values must always be compared to those of 17 beta-estradiol which, if they are low below a threshold value of 20 pg/ml, confirm that the oocyte reserve is exhausted.

Even if the hormonal dosages in this transition phase should be interpreted as a photograph of the instant in which they are dosed, but these values can fluctuate considerably over time and return to normal values followed by the appearance of almost regular cycles.

The values of 17betaestradiol were not determined in the manuscript.

So I would like to know what kind of reasoning the authors used in their methodology.

Author Response

We appreciate your careful and accurate descriptions of ways in which menopause can be biochemically diagnosed in clinical situations using FSH and estradiol. However, in our aim to answer the research question: “Does the age at menopause differ in women living with HIV and control women”, we selected a simple biochemical research measure tool in conjunction with the World Health Organization definition of menopause (described in lines 58-60). Based on peer reviewed, published HIV literature (in a high-ranking gynecology journal), we used an FSH value of ≥25 IU/ml as our biochemical criterion for menopause in addition to the absence of menses for at least one year (Cejtin HE, Kalinowski A, Bacchetti P, et al. Effects of human immunodeficiency virus on protracted amenorrhea and ovarian dysfunction. Obstet Gynecol 2006;108(6):1423-1431. (108/6/1423 pii ;10.1097/01.AOG.0000245442.29969.5c doi). Estradiol alone, as you rightly point out, is not sufficiently reliable since it is highly variable. However, FSH is less variable, especially as number of months since last menstrual flow increases. Similarly, the addition of estradiol would not aid our ability to distinguish menopause and hypothalamic amenorrhea, which was a primary aim of this work. In addition to this, we needed to limit both the blood drawn and the costs given research funding limitations.

Our results showed little difference between self-reported amenorrhea for one year (a clinically accepted diagnosis of menopause) and this biochemical criterion (FSH levels).  We used the same methods in both women living with HIV and the control groups and showed no difference. This was the primary outcome of our study.

Therefore, although it would have been interesting to have measured estradiol in all women in order to confirm a menopausal state, we believe it is not necessary in this research study. We have added a line to the discussion briefly describing when it would be clinically useful to measure estradiol, along with other hormone tests:

Lines 337-340: These results emphasize the importance of including biochemical tests in women aged <40-45 years when clinically indicated (i.e., when menstrual periods have ceased for greater than one year), including FSH, luteinizing hormone, beta-hCG, prolactin, estradiol, and thyroid stimulating hormone. In such cases, biochemical testing may identify reversible causes of amenorrhea.

We concur that it would be useful to have done two measures of FSH at least one month apart. However, this introduces complex logistics into a study in women who have little available time and energy to contribute, having a significant and large detrimental effect on sample size, and would also have increased finite research costs. We previously address the limitations of cross-sectional data in lines 363-367.

The phenomenon of prolonged amenorrhea in women living with HIV is something with which we are very familiar and have studied (King EM, Albert AY, Murray MCM. HIV and amenorrhea: a meta-analysis. AIDS 2019;33(3):483-491. DOI: 10.1097/QAD.0000000000002084). It especially happens in women who use opioids and experience subsequent weight loss. For that reason, we have performed a sensitivity analysis with these women (reported in lines 273-284).

Thank you for your thoughtful evaluation of our manuscript. We trust that this explanation will allow you to accept our research results.  
